# Estonian Named Entity Recognition: New Datasets and Models

**Kairit Sirts**
Institute of Computer Science
University of Tartu
`sirts@ut.ee`

## Abstract

This paper presents the annotation process of two Estonian named entity recognition (NER) datasets, involving the creation of annotation guidelines for labelling eleven different types of entities. In addition to the commonly annotated entities such as person names, organization names, and locations, the annotation scheme encompasses geopolitical entities, product names, titles/roles, events, dates, times, monetary values, and percents. The annotation was performed on two datasets, one involving reannotating an existing NER dataset primarily composed of news texts and the other incorporating new texts from news and social media domains. Transformer-based models were trained on these annotated datasets to establish baseline predictive performance. Our findings indicate that the best results were achieved by training a single model on the combined dataset, suggesting that the domain differences between the datasets are relatively small.

## 1 Introduction

Named entity recognition (NER) is a practical natural language processing (NLP) task that involves identifying and extracting named entities from texts, such as person names, organization names, locations, and other types of entities. NER is widely used in various downstream applications, such as document anonymisation and text categorisation. Typically, modern NER systems are trained as supervised tagging models, where annotated training data is utilised for training models to identify and tag text spans that correspond to named entities.

For the Estonian language, prior endeavors to develop NER systems have involved the creation of an annotated dataset labelled with person, organisation, and location names (Tkachenko et al., 2013). This dataset has been utilised for training CRF- and transformer-based NER models (Tkachenko et al., 2013; Kittask et al., 2020; Tanvir et al., 2021). In addition to these efforts, a dataset in a different domain, 19th-century parish court records, was recently annotated with named entities (Orasmaa et al., 2022).

This paper describes the efforts to augment further the development of general-purpose named NER systems for the Estonian language. The primary focus of this study is annotating additional Estonian texts with named entities, utilising a newly developed rich annotation scheme. Two annotated datasets were created as part of this effort. Firstly, the existing NER dataset (Tkachenko et al., 2013) was reannotated using the new annotation scheme. Secondly, approximately 130K tokens of new texts, predominantly sourced from news portals and social media, were annotated to create a new dataset. These annotations serve to expand the availability of annotated data for training and evaluating NER models in the Estonian language.

The second part of this paper delves into the experimental results obtained from training predictive BERT-based models on the annotated datasets. The primary objectives of these experiments were to establish the baseline performance of various entity types of the newly developed annotation scheme and to explore the optimal utilisation of the two datasets, which stem from slightly distinct domains. The findings revealed that the baseline performance on the newly annotated dataset was slightly lower than the less richly annotated Estonian NER dataset, indicating that the new annotations may possess some noise while also being richer and more intricate. Moreover, the study revealed that the domains of the two datasets were similar enough such that a model

trained on the combined dataset exhibited comparable or even superior performance compared to models trained on each dataset separately.

In short, our paper makes two key contributions:

1. The introduction of two novel Estonian NER datasets that are annotated with a comprehensive set of entities, enriching the available resources for NER research in Estonian;

2. An evaluation of the performance of BERT-based models on the newly annotated datasets, providing baseline assessments for these datasets.

## 2 Dataset Creation

This section describes the process of creating the two labelled NER datasets for Estonian.[1]

### 2.1 Data Sources

The first dataset, referred to as the Main NER dataset in our study, is a reannotation of the existing Estonian NER dataset (Tkachenko et al., 2013). This dataset comprises approximately 220K words of news texts and exhibits a homogeneous domain. Notably, previous studies have identified errors in the annotations of this dataset (Tanvir et al., 2021), which motivated us to undertake its reannotation.

The second dataset, referred to as the New NER dataset in our study, is newly created. We aimed to select approximately 130K tokens from news and social media domains, with around 100K tokens from the news domain and 30K tokens from the social media domain. To obtain the texts, we sampled from the Estonian Web Corpus 2017 (Jakubíček et al., 2013), utilizing metadata such as URL and web page title for text selection. For news sources, we identified URLs and titles associated with major Estonian news sites such as *Postimees*, *EPL*, *ERR*, and *Delfi*. For social media texts, we searched for keywords indicative of well-known blogging and forum platforms such as *blogspot* and *foorum*.

### 2.2 Annotation Guidelines

We devised annotation guidelines to label the data, aiming to adopt a more comprehensive set of labels beyond the commonly used person, organisation, and location names.[2] We decided to differentiate between geopolitical entities and geographical locations. Following similar works in Finnish (Ruokolainen et al., 2020), we introduced labels for events, products, and dates. Furthermore, we included titles, times, monetary values, and percentages. The annotation guidelines included a brief description for each entity, as used during the annotation process, which was as follows:

- Persons (PER): This includes names referring to all kinds of real and fictional persons.

- Organizations (ORG): This includes all kinds of clearly and unambiguously identifiable organizations, for example, companies and similar commercial institutions as well as administrative bodies.

- Locations (LOC): This includes all geographical locations not associated with a specific political organization such as GPEs.

- Geopolitical entities (GPE): This includes all geographic locations associated with a political organization, such as countries, cities, and empires.

- Titles (TITLE): This includes job titles, positions, scientific degrees, etc. Only those titles should be annotated where a specific person behind the title can be identified based on the preceding text. The personal name immediately following the title is not part of the TITLE. If the ORG tag precedes the title, only the job title must be marked with the TITLE, not the words in the ORG.

- Products (PROD): This includes all identifiable products, objects, works, etc., by name.

- Events (EVENT): This includes events with a specific name.

- Dates (DATE): This includes time expressions, both in day/month/year type, e.g., "October 3rd", "in 2020", "2019", "in September", as well as general expressions ("yesterday", "last month", "next year") if the expression has a clear referent. The criterion is that based on the expression, it must be possible to determine a specific point in time, i.e., a

---

[1]The annotated datasets are available:
https://github.com/TartuNLP/EstNER
https://github.com/TartuNLP/EstNER_new

[2]The annotation guidelines in Estonian are available upon request.

specific year, month, or day. Thus, vague expressions such as "a few years from now", "a few months ago" are not suitable, but more specific expressions such as "five years later", "three months ago", or "the day before yesterday" are suitable.

- Times (TIME): This includes time expressions that refer to an entity smaller than a day: times and parts of a day with a referent (analogous to DATE entities). General expressions without a referent are not marked. Durations are also not marked.

- Monetary values (MONEY): This includes expressions that refer to specific currencies and amounts in those currencies.

- Percentages (PERCENT): This includes entities expressing percentages. A percentage can be expressed both with a percentage mark (%) or verbally.

### 2.3 Nested Entities

Similar to Ruokolainen et al. (2020), we incorporated nested entities into our annotation schema. For instance, an example of a nested entity would be "New York City Government", where the ORG entity ORG encompasses the nested GPE entity "New York". We set a limit of up to three levels of nesting. However, we restricted the annotation of nested entities of the same type, except for ORG. For instance, if "The Republic of Ireland" was annotated as GPE, further annotation of "Ireland" as a nested GPE was not permitted. Nevertheless, in cases such as "The UN Department of Economic and Social Affairs" labelled as ORG, the word token "The UN" would be allowed to be annotated as a nested ORG.

### 2.4 Annotation Process

The process of annotation was carried out separately for both datasets. For the Main NER dataset, three annotators, who were graduate students in general or computational linguistics, were recruited. All annotators were native speakers of Estonian. Each annotator independently labelled the dataset based on the provided guidelines. Two annotators completed annotations for the entire dataset, while one annotator completed most of the annotations, with a few documents remaining. The annotation of the Main NER dataset was con-

ducted using Label Studio, a freely available open-source platform for data annotation.

A total of twelve annotators were involved in annotating the New NER dataset. Two annotators completed the entire annotation process. One of them was an undergraduate linguistic student, while the other was a graduate student in computer science with an undergraduate degree in linguistics. The remaining ten annotators participated in a graduate-level NLP course, and each annotated approximately 12K word tokens as part of their coursework. All annotators were native speakers of Estonian. All annotators worked independently, without access to each other's work, adhering to the provided annotation guidelines. As a result, each text in the New NER dataset received three independent annotations. The annotation of the New NER dataset was performed using DataTurks, an annotation platform currently non-existent.

### 2.5 Label Harmonisation

Harmonising the annotations in the New NER dataset involved both automatic and manual approaches. Initially, automatic harmonisation was applied based on the following principle. If annotators A and B had agreed on a particular annotation, but annotator C had not provided any annotation, the final label was set to the annotation agreed upon by A and B. Subsequently, the entire corpus was manually reviewed by two individuals, one of whom was the original annotator A, and the other was the author of this paper. Through discussion and deliberation, the labels were disambiguated. In most cases, the final label chosen was the one that at least two annotators had selected. However, in some instances, the label was changed entirely, or a completely new span of words was annotated as an entity based on mutual agreement.

The disambiguation of annotations in the Main NER dataset was carried out automatically. As per the automatic procedure, a word span was labelled as an entity if it had been marked as such by at least two annotators and they had used the same tag for that entity.

### 2.6 Inter-Annotator Agreement

In order to evaluate the reliability of the annotations, inter-annotator agreements were computed for the Main NER dataset, as shown in Table 1. Fleiss' kappa, an extension of Cohen's kappa to

|  | 1st level | 2nd level | 3rd level |
|---|---|---|---|
|  | 0.65 | 0.23 | -0.16 |
| PER | 0.95 | 0.27 | 0.66 |
| ORG | 0.76 | 0.33 | 0.19 |
| LOC | 0.65 | 0.35 | 0.18 |
| GPE | 0.84 | 0.47 | -0.08 |
| TITLE | 0.63 | 0.21 | 0.00 |
| PROD | 0.48 | 0.02 | – |
| EVENT | 0.43 | 0.53 | – |
| DATE | 0.72 | 0.06 | – |
| TIME | 0.53 | 0.00 | – |
| MONEY | 0.78 | 0.00 | – |
| PERCENT | 0.90 | – | – |

Table 1: Inter-annotator agreement of the Main NER dataset, measured with the Fleiss $\kappa$.

accommodate more than two annotators, was computed following the procedure outlined by Ruokolainen et al. (2020). Each entity occurrence in the text was treated as an instance of the positive class, and the exact match of annotations between annotators was checked for each entity. If annotators had marked the same entity with the same label, it was recorded as an instance of the positive class; otherwise, it was recorded as an instance of the negative class.

The inter-annotator agreement for the 1st level entities was found to be in the range of substantial agreement. However, in contrast, the annotations for the second and third levels showed lower agreement, as indicated by Fleiss' kappa's low or even negative values. Specifically, person names, geopolitical entities, and percentages achieved almost perfect agreement ($\kappa > 0.8$) at the first level. Most other entity types showed substantial agreement ($\kappa > 0.6$). The lowest agreement scores were observed for products and events, which still obtained moderate agreement ($\kappa > 0.4$).

### 2.7 Final Datasets

Following the label harmonisation process, the resulting datasets were divided into the train, validation, and test splits. These datasets and the prepared splits will be made available for future comparisons of developed models. Table 2 presents the final datasets' statistics.

The Main NER dataset was previously annotated with only three entity types: PER, ORG, and LOC, as reported by Tkachenko et al. (2013).

Among these, PER and ORG labels remain the most frequently occurring ones in the dataset. However, there have been changes in the annotation guidelines, resulting in most LOC annotations being replaced with GPE. Additionally, the Main NER dataset contains a relatively large number of titles, dates, and products. On the other hand, the occurrence of event entities is comparatively low in this dataset.

Similar trends in entity prevalence can be observed in the New NER dataset. PER, ORG, and GPE entities remain the most frequent, followed by a relatively large number of titles, dates, and products. Notably, the New NER dataset contains a higher occurrence of EVENT entities compared to the Main NER dataset. However, TIME, PERCENT, and MONEY entities are less frequent in the New NER dataset.

## 3 Experiments

We had two primary goals when conducting the experiments. The first goal was to establish the baseline performance on both the Main NER and New NER datasets. While several previous studies have reported results on the old annotations of the Main NER dataset, the new annotations we used in our study are more comprehensive and were collected independently without reference to the old annotations. Therefore, the baseline performance of the Main NER dataset with the new annotations may differ. Similarly, as the New NER dataset contains new material, it is crucial to evaluate its baseline performance as well.

The second goal of our study was to investigate potential domain differences between the two datasets. Specifically, the average document length in the New NER dataset was more than three times higher than that of the Main NER dataset. Also, the New NER dataset contains at least 30K tokens from the social media domain. Moreover, the news part of the New NER dataset documents was not limited to formal news texts but also included less formal opinion pieces. Hence, our objective was to determine the optimal approach for utilising these datasets, namely whether training separate models for each dataset would be more effective or if combining the data and training a single model would yield better results.

We opted to utilise only the first-level annotations for training our models. This decision was

|  | **Main NER dataset** | | | | **New NER dataset** | | | |
|  | **Train** | **Val** | **Test** | **Total** | **Train** | **Val** | **Test** | **Total** |
| Documents | 525 | 18 | 39 | 582 | 78 | 16 | 15 | 109 |
| Sentences | 9965 | 2415 | 1907 | 14287 | 7001 | 882 | 890 | 8773 |
| Tokens | 155983 | 32890 | 28370 | 217243 | 111858 | 13130 | 14686 | 139674 |
| 1st lvl entities | 14944 | 2808 | 2522 | 20274 | 8078 | 541 | 1002 | 9594 |
| 2nd lvl entities | 987 | 223 | 122 | 1332 | 571 | 44 | 59 | 674 |
| 3rd lvl entities | 40 | 14 | 4 | 58 | 27 | 0 | 1 | 28 |
| PER | 3563 | 642 | 722 | 4927 | 2601 | 109 | 299 | 3009 |
| ORG | 3215 | 504 | 541 | 4260 | 1177 | 85 | 150 | 1412 |
| LOC | 328 | 118 | 61 | 507 | 449 | 31 | 35 | 515 |
| GPE | 3377 | 714 | 479 | 4570 | 1253 | 129 | 231 | 1613 |
| TITLE | 1302 | 171 | 209 | 1682 | 702 | 19 | 59 | 772 |
| PROD | 874 | 161 | 66 | 1101 | 624 | 60 | 117 | 801 |
| EVENT | 56 | 13 | 17 | 86 | 230 | 15 | 26 | 271 |
| DATE | 1346 | 308 | 186 | 1840 | 746 | 64 | 77 | 887 |
| TIME | 456 | 39 | 30 | 525 | 103 | 6 | 6 | 115 |
| PERCENT | 137 | 62 | 58 | 257 | 75 | 11 | 1 | 87 |
| MONEY | 291 | 76 | 153 | 520 | 118 | 12 | 1 | 131 |

Table 2: Statistics of the two new Estonian NER datasets.

based on the finding that much fewer entities were labelled at the second and third levels, as evidenced by the statistics presented in Table 2. Furthermore, the inter-annotator agreements for the second and third-level entities were found to be lacking, as illustrated in Table 1. Hence, we focused solely on the first-level annotations to ensure a more reliable and consistent training process.

## 4 Model

We employed a transformer-based token classification model for our experiments, adopting the commonly-used BIO format for entity labelling. In this format, the B-tag indicates the start of an entity, the I-tag denotes the continuation of an entity, and the O-tag is assigned to word tokens that do not belong to any named entity. The TokenClassification implementation from the Huggingface `transformers` library (Wolf et al., 2020) was utilised for this purpose. As our base model, we used the EstBERT model with a sequence length of 128[3] (Tanvir et al., 2021), which was fine-tuned on the NER datasets.

In our experiments, we kept the batch size fixed

at 16 and utilised the Adam optimiser with betas set to 0.9 and 0.98 and an epsilon value of 1e-6. The models were trained for a maximum of 150 epochs, with early stopping implemented if the overall F1-score on the validation set did not improve for 20 consecutive epochs by more than 0.0001 F1-score points. We used the `seqeval` package (Nakayama, 2018) for evaluations during training and final testing. The learning rate was optimised on the validation set using a grid of values 5e-6, 1e-5, 3e-5, 5e-5, 1e-4. Each model was trained ten times with different random seeds to account for randomness, and the mean values with standard deviations are reported.

## 5 Results

We first trained and evaluated models separately on both datasets to assess their overall modeling performance. Then, we trained a joint model using data from both datasets and compared its performance on the evaluation sets of both datasets. This allowed us to evaluate the effectiveness of using a combined dataset compared to training on each dataset separately.

---

[3]https://huggingface.co/tartuNLP/EstBERT

| | | **Reannoated Main NER** | | | | **New NER** | | |
|---|---|---|---|---|---|---|---|---|
| | # | Precision | Recall | F1-score | # | Precision | Recall | F1-score |
| PER | 642 | .827 (.012) | .871 (.009) | .848 (.005) | 109 | .809 (.044) | .816 (.023) | .811 (.019) |
| ORG | 504 | .654 (.016) | .666 (.014) | .660 (.013) | 85 | .580 (.027) | .585 (.052) | .581 (.024) |
| LOC | 118 | .643 (.036) | .478 (.028) | .547 (.016) | 31 | .600 (.065) | .560 (.060) | .576 (.044) |
| GPE | 714 | .821 (.012) | .831 (.021) | .826 (.008) | 129 | .900 (.017) | .879 (.030) | .889 (.014) |
| TITLE | 171 | .676 (.023) | .814 (.014) | .739 (.011) | 19 | .750 (.062) | .718 (.064) | .731 (.048) |
| PROD | 161 | .572 (.033) | .628 (.026) | .598 (.024) | 60 | .509 (.043) | .474 (.052) | .488 (.029) |
| EVENT | 13 | .069 (.029) | .077 (.034) | .072 (.031) | 16 | .518 (.104) | .558 (.104) | .525 (.070) |
| DATE | 308 | .682 (.020) | .720 (.017) | .700 (.007) | 64 | .816 (.027) | .824 (.024) | .820 (.021) |
| TIME | 39 | .553 (.066) | .555 (.045) | .553 (.053) | 6 | .812 (.041) | .788 (.108) | .797 (.074) |
| PERCENT | 62 | .985 (.016) | .867 (.032) | .922 (.019) | 11 | .895 (.126) | 1 (–) | .940 (.074) |
| MONEY | 76 | .636 (.040) | .568 (.030) | .600 (.030) | 12 | .659 (.085) | .742 (.126) | .693 (.083) |
| Overall | 2571 | .737 (.010) | .757 (.009) | .747 (.004) | 497 | .736 (.014) | .734 (.017) | .735 (.006) |

Table 3: Predictive performance of models trained on both two datasets, evaluated on the respective validation set.

## 5.1 Separate Models

The results of the experiments with separate models, evaluated on the respective validation sets, are reported in Table 3. The overall performance, as indicated in the bottom row of the table, is similar for both datasets, suggesting that the annotation and modeling difficulty is comparable in the two datasets.

The entities that were most accurately predicted in both datasets are PER, GPE, and PERCENT. Conversely, the lowest accuracy was observed when predicting LOC, EVENT, and TIME for the reannotated Main NER dataset, and LOC, EVENT, and PROD for the New NER dataset. Predicting EVENT names is particularly challenging in the Main NER dataset, likely due to the limited number of instances (only 56) in the respective training set.

| | Precision | Recall | F1-score |
|---|---|---|---|
| PER | .948 | .958 | .953 |
| ORG | .784 | .826 | .805 |
| LOC | .899 | .914 | .907 |
| Overall | .891 | .912 | .901 |

Table 4: Results of the old annotations of the Main NER test set. Adapted from Table 11 (Tanvir et al., 2021).

A comparison of the results between the Reannotated Main dataset and the previous annotations of the Main NER dataset (refer to Table 4, sourced from Tanvir et al. (2021), Table 11) reveals that the performance on all three entities (PER, ORG, LOC) used in the old annotations has declined. It should be noted that the modeling results are not directly comparable, as Table 3 presents validation set results while Table 4 presents test set results. However, the differences in performance suggest that the new annotation might be more complex for the models to learn.

## 5.2 Joint Model

The joint model is trained using the combined train sets of the Main NER and New NER datasets. Table 5 presents the F1-scores of the joint model on the merged validation set and on the validation sets of both datasets individually. Notably, the overall F1-scores of the joint model are slightly higher than the F1-scores of the separate models (0.766 vs. 0.747 for the Main dataset and 0.752 vs. 0.735 for the New dataset), as evident from the bottom row of Table 3.

Figure 1 presents a detailed entity-level comparison of the joint and separate models on their respective validation sets. Specifically, Figure 1a illustrates the comparison on the validation set of the Main NER dataset. The results reveal that the joint model performs similarly or better than the separate models across most entities, except for the TIME entity, which already had low performance in the Main dataset and further decreases with the joint model from 0.553 to 0.433. Con-

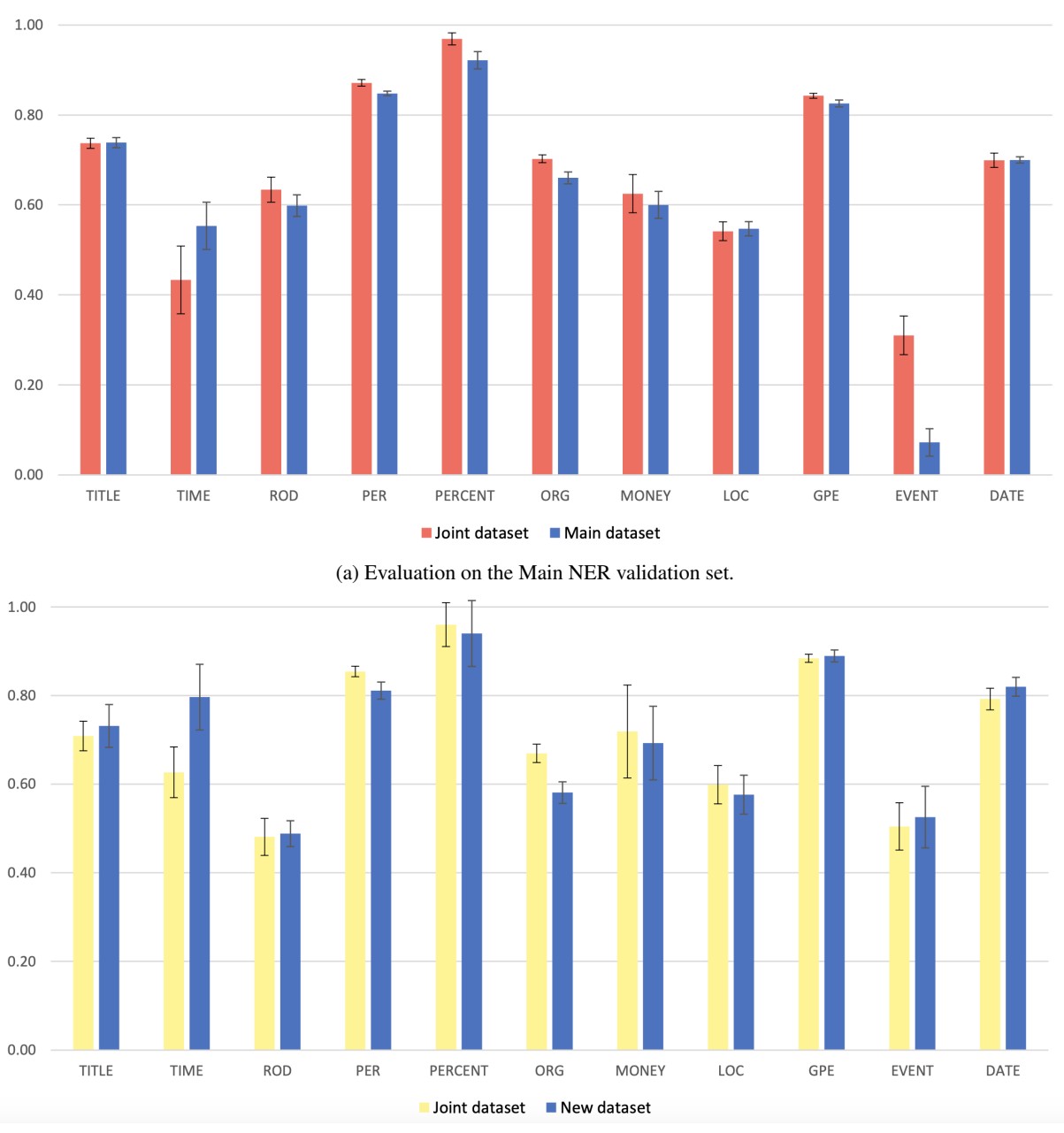

(a) Evaluation on the Main NER validation set.

(b) Evaluation on the New NER validation set.

Figure 1: An entity-level comparison of the joint model against models trained on each dataset separately.

|  | Main+New Val F1 | Main Val F1 | New Val F1 | Main+New Test F1 | Main+New Test Prec | Rec | F1 |
|---|---|---|---|---|---|---|---|
| PER | .868 (.007) | .872 (.008) | .854 (.012) | .879 (.007) | .840 | .927 | .882 |
| ORG | .690 (.010) | .702 (.009) | .669 (.021) | .700 (.016) | .698 | .693 | .696 |
| LOC | .549 (.019) | .541 (.021) | .599 (.043) | .526 (.025) | .478 | .563 | .517 |
| GPE | .849 (.005) | .843 (.005) | .884 (.009) | .826 (.004) | .827 | .830 | .828 |
| TITLE | .733 (.013) | .737 (.011) | .709 (.034) | .777 (.017) | .788 | .758 | .773 |
| PROD | .598 (.018) | .634 (.028) | .481 (.042) | .568 (.020) | .576 | .579 | .578 |
| EVENT | .370 (.053) | .310 (.043) | .504 (.053) | .264 (.034) | .306 | .256 | .278 |
| DATE | .708 (.013) | .699 (.016) | .792 (.024) | .740 (.010) | .727 | .768 | .747 |
| TIME | .451 (.065) | .433 (.075) | .627 (.057) | .463 (.043) | .548 | .472 | .507 |
| PERCENT | .969 (.019) | .969 (.013) | .960 (.049) | .958 (.013) | .967 | .983 | .975 |
| MONEY | .622 (.032) | .625 (.042) | .719 (.105) | .699 (.014) | .789 | .614 | .690 |
| Overall | .761 (.004) | .766 (.002) | .752 (.010) | .773 ( .006) | .766 | .783 | .774 |

Table 5: Evaluations of the joint model trained on the combined train sets of both datasets. Left block: F1-scores on the different portions of the validation sets. Middle block: F1-scores on the combined test set. Right block: test scores of the best-performing joint model.

versely, the prediction accuracy of the EVENT entity, while remaining relatively low, notably improves from 0.072 to 0.310 with the joint model.

Upon comparing the results of the joint and separate models on the New NER dataset (refer to Figure 1b), we observe that the joint model performs similarly or better on certain entity types, including PER, ORG, GPE, LOC, PROD, PERCENT, and MONEY while exhibiting slightly lower performance on the remaining entities. Notably, the TIME entity experiences the most significant drop in performance, declining from 0.797 to 0.627 with the joint model.

In summary, our findings support using a joint model instead of two separate models. While there may be a slight drop in prediction performance for certain entities, particularly in the New NER dataset, the overall F1-score on the validation sets of both datasets is higher with the joint model compared to the separate models. As a result, we proceed with the joint model for the final evaluations on the test set.

### 5.3 Test Results

The test results of the joint model on the combined test set can be found in the fourth column of Table 5. The overall F1-score is slightly higher on the test set than on the validation set. Specifically, for certain entities such as PER, ORG, TITLE, DATE, TIME, and MONEY, the test F1-score is higher than the validation F1-score, while

it is slightly lower for others. Notably, the EVENT entity experiences the most significant drop in performance, with the test F1-score declining from 0.370 to 0.264.

All the results mentioned above were presented as averages across ten different runs. Additionally, we selected a joint model with the highest overall validation F1-score to make it publicly available. The test scores of this chosen model are provided in the right-most block of Table 5. The overall F1-score of this best model is in line with the mean F1-score, indicating that it was not the model with the highest F1-score on the test set. However, due to the small standard deviations observed, the results of all models are within a close range; the highest F1-score achieved on the test set is 0.785.[4]

## 6 Discussion

This study marks the first endeavor to annotate a more comprehensive set of entities beyond the commonly annotated person, organization, and location names in the Estonian language. The inter-annotator agreement results indicate that the annotators consistently labelled certain entities, such as PER, GPE, and PERCENT, while the reliability was lower for other entities. In particular, the EVENT entity had the lowest inter-annotator agreement. An in-depth analysis of inconsistencies in annotation, both in EVENT and other en-

---

[4]The best joint model is available: `https://huggingface.co/tartuNLP/EstBERT_NER_v2`

tities, could be conducted as a follow-up work to identify the sources of confusion and enhance the annotation guidelines.

In line with previous efforts in other languages, such as Finnish, we opted to annotate nested entities by permitting up to three levels of nesting. However, upon analysing the data statistics, it was revealed that only a few entities were annotated on the third level. Additionally, even though many entities were labelled on the second level, their reliability, as evidenced by inter-annotator agreements, was not deemed sufficiently high. Hence, utilising these labels for training predictive models may not yield productive results.

In this study, we obtained three sets of annotations for both datasets, enabling us to assess the variability in the annotations. However, it is essential to acknowledge that the choice of annotators may have introduced limitations to the annotation process. For the Main NER dataset, all annotators were linguistic students, which provided expertise and interest in the annotation task, as intended. However, this uniformity in the background may have resulted in limitations in the recall of entity annotations, as noted in previous research (Derczynski et al., 2016). On the other hand, the annotators for the New NER dataset were more diverse, including computer science students. Nevertheless, since the task was part of their coursework, their motivation and interest in the annotation task might not have been as high.

Our experimental results with the BERT-based model indicate that although there may be a domain shift between the two datasets at the entity level for certain entities, training a single joint model on both datasets seems justified. It is important to note that our models based on EstBERT are only baselines, and as demonstrated in previous studies (Kittask et al., 2020; Tanvir et al., 2021), utilising other base models such as Estonian WikiBERT (Pyysalo et al., 2021) or XLM-RoBERTa could potentially yield higher performance results.

## 7    Conclusions

We provided a detailed overview of the annotation process for two Estonian NER datasets, annotated with a comprehensive annotation scheme encompassing eleven distinct entity types. Additionally, the datasets included nested annotations of up to three levels, although the reliability of the nested annotations was found to be less consistent compared to the first-level entities. In order to establish baseline predictive accuracy, we conducted experiments with two modeling scenarios on these newly annotated datasets. This involved training two separate models, one for each dataset and a joint model on the combined dataset. Our findings revealed that the joint model outperformed the separate models, except for a few entity types, indicating that the domain differences between the datasets are relatively minimal. As such, we recommend utilising these two datasets jointly as a single, more diverse dataset for NER training purposes.

## Acknowledgments

We thank Laura-Katrin Leman, Chenghan Chung and Claudia Kittask for their contributions in this work, and all data annotators. This research was supported by the Estonian Research Council Grant PSG721 and by the Estonian Language Technology Grant EKTB11.

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
