# OpenReview forum: "Estonian Named Entity Recognition: New Datasets and Models"
_NoDaLiDa/2023/Conference — NoDaLiDa 2023_

### Official Review · Reviewer_VAeW · 2023-03-14
**NER dataset for Estonian and baseline benchmark results**

**Rating:** 7
**Confidence:** 4

**Review:**

The paper describes 1) the annotation of a new Estonian named entity recognition dataset, and 2) benchmarking results for a number of baselines using this dataset.

This is a straightforward resource paper. Most of the text describes the annotation procedure for the dataset. The inventory of NER labels are enumerated and some demarcation criteria are detailed. The annotation is multi-level, so that entities can potentionally be nested. The steps of annotation, quality checks, and harmonization are described in the paper.

The last few sections of the paper describe some results in applying a token-classification transformer to the new dataset. An Estonian BERT model is fine-tuned. The paper discusses some variations of this model. There are no major differences in performance between the variations. There are also some analysis of the cross-domain behavior of these models; the drop in accuracy between domains is quite small.



**Paper Type:**

Long paper

---

### Official Review · Reviewer_YPyY · 2023-03-16
**Bit long but fine enough**

**Rating:** 6
**Confidence:** 4

**Review:**

This paper is exactly what one expects given its title. That is good news.

Parts missing:
* The annotators are insufficiently described. Please include an NLP Data Statement style report. There is ample room, esp. given how colossal Fig 1 is.
* Three grad students from the same discipline is very weak model for NER annotation. See e.g. Table 3 in https://aclanthology.org/C16-1111/ , where a homogeneous annotator population achieve high agreement but perform poorly, whereas a heterogeneous annotator population notices different things and overall annotates a lot more of the things that should be annotated.
* Where do we get the data? Especially, under what license? This must be explained in the paper prose - I have assumed the worst, that it is not yet available and will have some tight license, in the absence of any promises otherwise
* How have you annotated fictional persons e.g. Harry Potter? What about contentious entities e.g. God? What about use of locations as organisations, e.g. "Barcelona lost to Tartu in penalties last night"? These questions are well-addressed in the field of NER but poorly addressed in this paper.
* Where do you say what model you use? I only found info about the tokeniser and optimiser settings. Without this being super clear, the paper is a reject, but I've assumed this is trivial to add for camera ready. If it's not in camera ready, I will try to find you at the conference and enter into a particularly slow and tedious discussion on the topic (no, only joking. but please include it! :) )

**Paper Type:**

Long paper

---

### Official Review · Reviewer_WWyN · 2023-03-17
**estonian NER**

**Rating:** 7
**Confidence:** 3

**Review:**

This paper presents a re-annotation and new annotation of two datasets (one of which had already been annotated for NER in the past). The paper also features some first number on NER detection using established methods to provide a baseline for the future.

The paper is well written and the content is sound. NoDaLiDa is exactly the right venue for such work.

A few comments:
- it is unfortunate that the pre-existing annotation has not been further compared to the new one (beyond what's in Sec. 2.6), including training/testing of the models on the old annotation. Are there more advantages beyond the finer granularity of the new annotation? In hindsight, does it help more to reannotate or just to newly annotate more material?
- regarding domain difference, have you tried informing the model about the domain, e.g. via a prepended marker-token and analyzed a possible influence on the performance?

**Paper Type:**

Long paper

---

### Decision · Program_Chairs · 2023-03-17

Accept